# Overexpression of *Isl1* under the *Pax2* Promoter, Leads to Impaired Sound Processing and Increased Inhibition in the Inferior Colliculus

**DOI:** 10.3390/ijms22094507

**Published:** 2021-04-26

**Authors:** Tetyana Chumak, Diana Tothova, Iva Filova, Zbynek Bures, Jiri Popelar, Gabriela Pavlinkova, Josef Syka

**Affiliations:** 1Institute of Experimental Medicine CAS, Vídenska 1083, 14220 Prague, Czech Republic; diana.tothova@iem.cas.cz (D.T.); zbynek.bures@iem.cas.cz (Z.B.); jiri.popelar@iem.cas.cz (J.P.); 2Institute of Biotechnology CAS, 25250 Vestec, Czech Republic; iva.filova@ibt.cas.cz; 3Department of Technical Studies, The College of Polytechnics, 58601 Jihlava, Czech Republic

**Keywords:** transcription factor ISL1, auditory system, sound processing, inferior colliculus, inhibition

## Abstract

The LIM homeodomain transcription factor ISL1 is essential for the different aspects of neuronal development and maintenance. In order to study the role of ISL1 in the auditory system, we generated a transgenic mouse (*Tg*) expressing *Isl1* under the *Pax2* promoter control. We previously reported a progressive age-related decline in hearing and abnormalities in the inner ear, medial olivocochlear system, and auditory midbrain of these *Tg* mice. In this study, we investigated how *Isl1* overexpression affects sound processing by the neurons of the inferior colliculus (IC). We recorded extracellular neuronal activity and analyzed the responses of IC neurons to broadband noise, clicks, pure tones, two-tone stimulation and frequency-modulated sounds. We found that *Tg* animals showed a higher inhibition as displayed by two-tone stimulation; they exhibited a wider dynamic range, lower spontaneous firing rate, longer first spike latency and, in the processing of frequency modulated sounds, showed a prevalence of high-frequency inhibition. Functional changes were accompanied by a decreased number of calretinin and parvalbumin positive neurons, and an increased expression of vesicular GABA/glycine transporter and calbindin in the IC of *Tg* mice, compared to wild type animals. The results further characterize abnormal sound processing in the IC of *Tg* mice and demonstrate that major changes occur on the side of inhibition.

## 1. Introduction

The inferior colliculus (IC) plays an important role in auditory processing and integrating information about the spectral, temporal, and spatial features of sound that has been preprocessed by multiple specialized brainstem networks [1,2]. The IC is morphologically divided into the core (central nucleus) and shell (dorsal and external nucleus) subdivisions. The central nucleus is tonotopically organized, receives ascending auditory inputs from the brainstem auditory nuclei, and projects to the thalamus, from where the auditory signal proceeds to the cortex. The IC shell plays a neuromodulatory role, not only receiving and processing ascending auditory information, but also descending projections from the auditory cortex (targeting mainly dorsal nucleus [3]), and inputs from the IC core and non-auditory structures (external nucleus [4]). IC neurons are represented by two major classes, excitatory glutamatergic neurons and inhibitory GABAergic neurons. GABAergic neurons comprise about 20–30% of the inhibitory neurons in the IC of rodents [5,6,7,8], and 15% in the human IC [9]. Despite the lower proportion of GABAergic cells in the IC, inhibition plays a critical role in shaping IC neuronal responses [10].

*Isl1* is a LIM-homeodomain transcription factor and is essential for normal embryonic development, since embryos lacking the *Isl1* gene die by embryonic day 10.5 (E10.5) [11]. Loss-of-function studies in mice have revealed that ISL1 has a crucial role in neuronal, pancreatic, cardiac, and eye development [12,13,14,15,16]. ISL1 acts in a context-dependent fashion, and has the unique potential to combinatorially interact with other transcription regulators in a homomeric or heteromeric fashion [17]. To further explore the function of ISL1, we generated a gain-of-function model with the transgenic expression of *Isl1* under *Pax2* regulatory sequences. *Pax2* is one of the earliest genes detectable in the prospective mid-hindbrain region (E7.5) [18], it is indispensable for the normal development of this brain region [19], and it is essential for inner ear development [20]. We hypothesized that by using an overexpression model of *Isl1* under the *Pax2* regulatory sequence, we can further explore the role of ISL1 in the auditory system. We previously showed that *Pax2-Isl1* transgenic mice (*Tg*) exhibit an altered development of the auditory system at multiple levels, affecting the inner ear, auditory brainstem [21], and midbrain [22]. These *Tg* mice demonstrated an early onset of the age-related hearing loss compared to littermate controls [21]. The accelerated age-related hearing loss was associated with a functional inefficiency of cochlear outer hair cells and the deterioration of the medial olivocochlear efferent system. Additionally, our behavior analyses show that *Pax2*-driven ISL1 overexpression alters GABA signaling, as indicated by the systemic responsiveness to picrotoxin, a non-competitive channel blocker for the γ-aminobutyric acid (GABA) receptor chloride channels [22]. 

In this study, we further explore the gain-of-function role of *Isl1* on acoustic signal processing and the functional and molecular properties of neurons in the IC of *Tg* mice. Altered development of the IC was manifested in its decreased size and in the reduced amplitude of late ABR waves. The decreased size of the IC in *Tg* mice did not affect the range of frequency representation. However, *Pax2*-driven ISL1 overexpression resulted in various deficits in sound processing in the IC, affecting the function of both low and high frequency neurons. Our results suggest that the defects of auditory processing in the IC of the *Tg* mice were primarily caused by impaired inhibitory signaling. 

## 2. Results

### 2.1. Young Tg Animals Exhibit Slightly Impaired High Frequency Hearing

Prior to the recording of neuronal activity in the IC, we performed a basic evaluation of hearing function via recording and analysis of DPOAEs and ABRs. DPOAEs reflect the motile function of the outer hair cells responsible for sound amplification in the inner ear, while ABRs represent summed neuronal activity in distinct nuclei along the ascending auditory pathway. The ABR in mice consists of five positive waves [23,24,25], of which wave IV represents activity of the lateral lemniscus and partially reflects the activity in the IC [26] and wave V originates mainly in the IC [25]. In mice, early waves I to III have usually the highest amplitude, while wave V is often unstable and tends to blend into the noise level [24]. Therefore, to evaluate the sound evoked response at the level of upper brainstem and midbrain in relation to the action potential in the auditory nerve, we analyzed the ratio of the ABR wave I to wave IV amplitude.

WT animals (*n* = 8) showed a normal amplitude and profile of DPOAEs, with a 40dB SPL peak at 18–20 kHz. The DPOAE amplitude in *Tg* animals (*n* = 9) was slightly but significantly decreased in the high frequency region (Figure 1A, interaction between Frequency and Genotype *p* = 0.02; repeated measures two-way ANOVA, Uncorrected Fisher’s LSD post-hoc test), confirming an impaired function of outer hair cells in the basal part of the cochlea, as previously reported [21]. The outer hair cell functional deficit was, however, not strong enough to significantly affect the hearing thresholds assessed using ABR recording, and no difference was shown between the *Tg* and WT animals (Figure 1B, repeated measures two-way ANOVA). The ABR wave analysis performed using click-evoked ABR response curves showed a significant increase of ABR wave I to wave IV amplitude ratio (Figure 1C, *p* = 0.047; unpaired *t*-test), due to the decreased amplitude of wave IV in the mutant mice compared to WT (Figure 1D), suggesting a sound processing deficiency in the auditory brainstem and midbrain and confirming our previous report [22].

### 2.2. The Inferior Colliculus Is Smaller in Tg Mice

The IC of *Tg* mice was significantly smaller than in WT animals (Figure 1E, WT: 2.29 ± 0.1 mm^2^, *Tg:* 1.55 ± 0.5 mm^2^), and this difference is often easily detectable by a simple visual inspection of the dorsal brain surface (Figure 2A,B), as well as in brain sections (Figure 2C,D). To characterize the response properties of the IC neurons of *Tg* and WT animals, we performed multiunit extracellular recordings of neuronal activity. Two animals with the most pronounced reduction of IC size were excluded from our study due to the inability to insert an electrode for neuronal response recording.

### 2.3. The Properties of IC Neuronal Responses Are Abnormal in Tg Mice

We subsequently investigated the tuning properties of IC neurons at various frequencies. In both the WT and *Tg* mice, we recorded and analyzed responses from neurons with characteristic frequencies from 4 to 35 kHz (Figure 3). Since the processing of sounds of high frequency could be affected, as suggested by impaired DPOAEs, the analyzed IC neurons were divided into two groups based on their neuronal characteristic frequency: low frequency neurons (LFN, 4–14 kHz, WT *n*= 284, *Tg n* = 414) and high frequency neurons (HFN, 15–35 kHz, WT *n* = 315, *Tg n* = 266). Neuronal thresholds corresponding to the lowest sound intensity detectable by neurons were determined using both tone- and BBN-evoked neuronal responses. Surprisingly, despite the deficit of DPOAEs in the high frequency region of the *Tg* cochlea, elevated sound thresholds in responses to both pure tones and BBN were only found in the LFN in the IC of the transgenic mice (tone-evoked responses: Figure 3A,B, WT: Mdn = 20.29 dB SPL, *Tg*: Mdn = 24.62 dB SPL, *p* < 0.0001, Mann-Whitney test; BBN-evoked responses: Figure 4A, WT Mdn = 21.27 dB SPL, *Tg* Mdn = 22.28 dB SPL, *p* < 0.0001, Mann-Whitney test), and not in the HFN (Figure 3A,C and Figure 4D, Mann-Whitney test). To assess the frequency selectivity, quality factor Q_10_ was compared in the IC neurons of the WT and *Tg* mice. We found no difference between the animal groups in the LFN frequency selectivity; in both groups the Q_10_ Mdn was 1.8 (Figure 3E). In contrast, the HFN of the *Tg* mice exhibited significantly impaired frequency selectivity, manifested in a decreased quality factor (Figure 3D,F, WT Mdn = 2.1, *Tg* Mdn = 1.9 dB SPL, *p* = 0.009, Mann-Whitney test).

Rate-intensity functions were analyzed using IC neuronal responses to the BBN stimulation of different intensity. In addition to the above-mentioned increased neuronal thresholds, the neurons in the *Tg* IC exhibited a wider dynamic range compared to the WT, and this difference was more pronounced for the LFN than for HFN (Figure 4B,E, LFN WT: Mdn = 30.91 dB, *Tg*: Mdn = 39.49, *p* < 0.0001, HFN: WT: Mdn = 32.41 dB, *Tg*: Mdn = 38.48, *p* = 0.002). No difference between the groups in maximum response was found in the LFN, while HFN showed a smaller maximum response in the *Tg* mice compared to the WT IC (Figure 4C,F, WT: Mdn = 13.11 spike/s, *Tg*: Mdn = 11.65, *p* = 0.032). 

The FSL was extracted from the post-stimulus time histograms of responses to BBN stimuli. Being a precise and stable function of a stimulus amplitude [27], the FSL increased with a decrease of stimulus intensity in both LFN and HFN, in the IC of both groups of animals (Figure 5A,D). *Tg* animals, however, already exhibited prolonged FSL in response to the highest recorded sound intensities, and this deficit increased gradually with decreasing sound level (repeated measures two-way ANOVA, main effect of genotype *p* < 0.0001, interaction between genotype and sound intensity *p* < 0.0001), reaching an approximate 2 ms delay compared to WT in response to the lowest analyzed intensity of 30 dB SPL (Figure 5A, LFN: Mean = 12.5 ms for WT and 14.56 for *Tg*, *p* < 0.0001; Figure 5D, HFN: Mean = 12.05 for WT and 14.34 for *Tg p* < 0.0001, repeated measures two-way ANOVA, Uncorrected Fisher’s LSD post-hoc test).

The spontaneous firing rate was extracted from the last 100 ms of the recorded activity to stimulation using low-level BBN stimuli. In both groups of animals, it ranged from 0 to approximately 200 spikes/s for the LFN and HFN. The median spontaneous activity, however, was lower in the LFN (Figure 5B, WT Mdn = 17.5 spike/s, *Tg* Mdn = 5 spike/s, *p* < 0.0001, Mann-Whitney test) and the HFN (Figure 5E, WT Mdn = 7.5 spike/s, *Tg* Mdn = 0 spike/s, *p* < 0.0001, Mann-Whitney test) in the IC of *Tg* mice compared to the WT, due to the higher number of neurons with zero spontaneous activity in both groups of transgenic IC neurons (Figure 5C,F).

We then tested the ability of neurons to follow and respond precisely to a train of click stimulations with an inter-click interval ranging from 2 to 100 ms. Both groups of IC neurons from the *Tg* and WT animals were able to follow click trains with inter-click intervals from 100 to 30 ms with a comparable fidelity (Figure 6A,B). However, the HFN in the transgenic IC exhibited a significant drop in synchronization ability in responses to click trains, with shorter inter-click intervals (from 20 to 5 ms) (Figure 6B). On the other hand, the LFN showed slightly higher vector strength values at 5 ms and 2 ms inter-click intervals (Figure 6A).

### 2.4. Inhibition Is Increased in the Inferior Colliculus of Mutant Mice

Two-tone stimulation was employed to assess the high- and low-frequency inhibition in the LFN and HFN of the WT and *Tg* mice. The strength of inhibition was expressed as a percent of the suppression of the firing rate in the low- and high-frequency sidebands enframing the tuning curve [28]. The LFN in the *Tg* IC showed an increased inhibitory strength on both, low (Figure 7A, WT Mdn = 45.19%, *Tg* Mdn = 56.52%, *p* < 0.01) and high (Figure 7B, WT Mdn = 58.93%, *Tg* Mdn = 71.81%, *p* < 0.001) frequency sides, while the inhibitory strength of the HFN was similar between the WT and *Tg* animals (Figure 7D,E).

Another indirect measure of inhibition in the auditory system is the DSI, which is calculated as the difference of spiking activity in response to the up- and downward FM stimulus [29,30]. The right shift of DSI suggests prevalence of high frequency inhibition over low frequency inhibition. In both groups of experimental animals, the DSI was shifted to the right, reflecting prevalence of high frequency inhibition. However, in both types of neurons, this shift was significantly more pronounced in the *Tg* animals (Figure 7C, LFN: Mdn = 0.08 for WT and 0.17 for *Tg*, *p* < 0.0001; Figure 7F, HFN: Mdn = 0.05 for WT and 0.16 for *Tg p* < 0.0001, Mann-Whitney test) suggesting even higher dominance of the high frequency inhibition in the *Tg* IC.

### 2.5. The Molecular Profile and Distribution of Neurons Are Altered in the Inferior Colliculus of Mutant Mice

Having recognized the major effects of the transgenic expression of *Isl1* on the frequency tuning, intensity, and temporal coding properties of IC neurons, we next wanted to establish the cellular and molecular differences between the *Tg* and WT IC. Calcium binding proteins, calbindin, calretinin, and parvalbumin, are abundantly expressed in the IC and their expression patterns define specific IC subdivisions: calbindin is mainly expressed in neurons in the dorsal nucleus particularly in the proximity to the commissure of the IC, calretinin in the dorsal nucleus, and parvalbumin throughout the IC [8,31]. In our study, we focused on two types of IC neurons, parvalbumin and calretinin immunolabeled neurons (Figure 8A,B). Calretinin^+^ neurons dominate in the dorsal half of the IC (Figure 8A), which contains low frequency neurons and is the most affected part of the IC in the *Tg* mouse. We found a reduced number of calretinin^+^ cells in both, the dorsal (DIC) and central (CIC) IC nuclei in the *Tg* brain compared to the WT (Figure 8C, *p* < 0.001, *t*-test). 

Parvalbumin immunolabeled cells have been shown to colocalize with GABA, and, thus, inhibitory cells in the IC [32]. The number of parvalbumin^+^ neurons was decreased in the CIC of the *Tg* compared to the WT animals (Figure 8C, *p* < 0.01, *t*-test). We also quantified the expression of mRNA of *Slc32a1* (encodes a transporter involved in GABA and glycine uptake into synaptic vesicles), and genes encoding calcium binding proteins (calbindin (*Calb1*), calretinin (*Calb2*), and parvalbumin (*Pvalb*)). The relative quantification of mRNA expression was performed in the whole IC using RT-qPCR. The expression of *Calb1* and *Slc32a1* was significantly increased, indicating changes in the molecular expression profile of IC neurons in the *Tg* mice, while we found no difference between the groups in the levels of mRNA encoding calretinin and parvalbumin (Figure 9, *p* < 0.01, *t*-test).

## 3. Discussion

In this study, we investigated how the overexpression of *Isl1* under *Pax2* regulatory sequences, associated with abnormalities in the development of the mid-hindbrain region, affects sound processing by IC neurons. We performed extracellular recording of neuronal activity in the IC of *Tg* and WT mice. Animals six to eight weeks old were used in the study to minimize the effects of premature loss of high frequency hearing, previously reported for these mutants [21]. Compared to WT littermates, *Tg* mice at this age already had decreased DPOAE amplitudes in the high frequency part of the cochlea, but without any significant shift of the hearing thresholds.

The decreased size of the IC in *Tg* mice did not affect the range of frequency representation. However, the mutant mice demonstrated various deficits in sound processing in the IC, affecting the function of both low and high frequency neurons, often the LFN to a higher extent. Our results suggest that the defects of auditory processing in the IC of the *Tg* mice were primarily caused by impaired inhibitory signaling. This agrees with our previous report of increased inhibition in the brain of the *Tg* mice resulting in hyperactivity, which was successfully reduced by the administration of a non-competitive channel blocker for the GABA receptor chloride channels [22]. Inhibition is an important phenomenon, shaping and limiting responses throughout the auditory pathway. In the IC, inhibition is of particular importance, as it plays a determinative role in neuronal selectivity to social vocalization [33], and binaural sound processing [34]. Using two-tone stimulation to uncover inhibitory areas in the IC neuronal receptive fields, we found a higher strength of low- and high-frequency sideband inhibition in the LFN in the *Tg* IC. Neuronal receptive fields in the IC are a result of the input from the lower auditory nuclei, and interplay of the intracollicular excitation and inhibition. Thus, the origin of sideband inhibition in the IC remains disputable. The two-tone stimulation paradigm [35], which we used to detect inhibitory sidebands and calculate inhibitory strength, involves the simultaneous presentation of two tones to the animal: one tone of variable frequency and intensity, together with another tone fixed at CF 10 dB above the threshold. This stimulation allows the discovery of inhibitory areas around the excitatory tuning curve, represented by a suppressed response to the “fixed tone” [28]. A similar setting, however, evokes two-tone suppression at the level of cochlear basilar membrane, which could potentially, together with inhibition in the lower auditory nuclei, contribute to the two-tone response areas in the IC neuronal response receptive fields. Multiple studies, however, argue with the extra-collicular origin of two-tone inhibition recorded in the IC. Thus, the presence of similar inhibitory patterns in response areas of neurons with spontaneous activity, in response to a single tone stimulation [36,37] and studies using GABA or glycine antagonists in the IC, [38,39,40] suggest prevalence of the intra-collicular inhibition in the IC tuning curves recorded in response to the two-tone stimulation. Interestingly, the increased sideband inhibition in the low frequency region of the IC could lead to increased hearing thresholds of the LFN responses to both pure tone and BBN stimulation, despite the DPOAE deficit in the high frequency part of the *Tg* cochlea. A similar effect of inhibition has been shown in the studies using GABA and glycine antagonists. For example, a bicuculline blocking side band inhibition caused 5–15 dB threshold reduction in 20% of IC neurons in guinea pigs [39], similar to the effect of the combination of bicuculline and strychnine in the Mexican-free tailed bat [33].

In contrast to the results of two-tone stimulation, the analyses of other indirect measures of inhibition, such as neuronal selectivity for FM sweep direction, FSL and spontaneous firing rate, showed similar effects in the LFN and HFN. Thus, the index of direction sensitivity to FM sweeps showed higher prevalence of the high frequency inhibition in both the LFN and HFN in the *Tg* IC, compared to those of the WT mice. FM sweeps are a universal component of animal vocalizations and the IC has been shown to be the first nucleus in the auditory pathway, in which neurons exhibit direction selectivity to FM sweeps as a result of interplay between excitatory and inhibitory synaptic inputs [29]. The direction selectivity to FM sweeps was shown to originate in the IC as an effect of the sideband inhibition [41]. Similarly, we found longer first spike latency in the *Tg* IC, compared to WT neuronal responses to sounds in all the tested sound pressure levels. This could be an effect of the increased inhibition, since a blockage of GABA_A_ receptor with bicuculline led to the shortening of the response latencies [42]. The decreased sound evoked firing rate and spontaneous activity rate in the IC of *Tg* mice compared to WT IC neurons could also be an effect of increased inhibition, as GABA_A_ receptor blockage leads to an increased discharge rate in the IC of the chinchilla [43]. Changes in the rate-intensity functions manifested in the increased dynamic range of *Tg* IC neurons and an increased response threshold in the LFN, also indicate an altered inhibition in the *Tg* IC. Therefore, intensity coding in the IC is dependent on the combination of the external sources and local colliculi circuits in which the inhibition plays a prominent role [44].

Electrophysiological evidence of increased inhibition was supported by the significant increase of expression of mRNA for vesicular GABA/glycine transporter *Slc32a1* and calcium binding protein, calbindin, in the IC. At the same time, similarly to the findings in the cerebellum [22], we found a significantly decreased number of calretinin expressing neurons in the CIC and DIC and parvalbumin^+^ neurons in the CIC of the *Tg* animals. Calcium binding proteins are expressed throughout the central nervous system and are essential for calcium homeostasis during cellular signaling, synaptic function, and neuronal plasticity. They frequently colocalize with GABA positive inhibitory interneurons. In the central nucleus of the IC, parvalbumin has been shown to label all GABA immune-positive cells, while solitary calretinin and calbindin positive cells did not colocalize with GABA or glutamate [32]. To our knowledge, no such information is available for the IC shell neurons. The role of these proteins in the auditory midbrain is not fully understood. The specificity of calcium binding proteins, as markers of particular anatomical and functional subsets of neurons, has been discussed [45,46]. Thus, the pattern of distribution of calretinin^+^ neurons throughout the auditory pathway nuclei, suggest their relation to the subset of neurons responsible for binaural hearing [47]. Whether binaural sound processing is affected in the *Tg* animals remains to be established.

In this study, we established multiple auditory processing deficits in the IC of *Tg* mice mainly conditioned by increased inhibition. Further studies employing these *Tg* mice may support deciphering the role of inhibition in sound processing, and the molecular changes associated with the overexpression of *Isl1* in neurons of the IC.

## 4. Materials and Methods

### 4.1. Animals

Heterozygous *Pax2-Isl1* transgenic mice (*Tg*) [21,22] and their wild type (WT) littermates of both sexes were used in the study. The experimental protocol was approved by the Animal Care and Use Committee of the Institute of Molecular Genetics, Czech Academy of Sciences. Animals were housed in a controlled environment (23 °C; 12-h light/dark cycle) with free access to water and standard chow diet.

All hearing tests were carried out on mice anesthetized via an intraperitoneal injection of a mixture of 35 mg/kg ketamine (Calypsol 50 mg/mL; Gedeon Richter, Budapest, Hungary), and 6 mg/kg of xylazine (Xylapan 20 mg/mL; Vetoquinol SA, Lure Cedex, France). The body temperature was maintained with a DC-powered electric temperature regulated pad. The recordings were carried out in a soundproof anechoic room.

### 4.2. Distortion Product Otoacoustic Emissions

Cubic (2 F1–F2) distortion product otoacoustic emissions (DPOAEs) over an F2 frequency range from 4 to 38 kHz, were recorded with a low-noise microphone system (Etymotic probe ER-10B+, Etymotic Research, Elk Grove Village, IL, USA). Acoustic stimuli (ratio F2/F1 = 1.21, F1 and F2 primary tone levels of L1/L2 = 70/60 dB) were presented to the ear canal with two custom-made piezoelectric stimulators connected to the probe with 10-cm-long silastic tubes. The signal from the microphone was analyzed by the TDT System III (RP2 processor, sampling rate 100 kHz) (Tucker Davis Technologies, Alachua, FL, USA) using custom-made Matlab software. DPOAEs were successively recorded in both ears of the animals at individual frequencies over the frequency range 4–38 kHz, with a resolution of four points per octave. The average values per group were calculated (mean ± SD) and presented as DP-grams.

### 4.3. Auditory Brainstem Responses

Auditory brainstem responses (ABRs) were measured using three needle electrodes placed subcutaneously on the mouse vertex (recording electrode), and on two sides of the animal’s neck (ground and reference electrodes). To evoke ABRs, the animals were exposed to click (angular pulse with alternating polarity, duration 0.1 ms, repetition rate of 11 Hz) or pure tone stimuli (3 ms duration, 1 ms rise/fall times, frequencies 2–40 kHz), of gradually decreasing sound pressure with 5 dB step. Acoustic stimuli were conveyed to the animal in free-field conditions via a two-way loudspeaker system (Jamo^®^ woofer (Glyngøre, Denmark), and a SEAS^®^ T25CF 002 tweeter (Oslo, Norway)), which was placed 70 cm in front of the animal’s head. The signal from an electrode was amplified and band-pass filtered over a range of 300 Hz to 3 kHz, processed with a TDT System III Pentusa Base Station (Tucker Davis Technologies, Alachua, FL, USA) and analyzed using BioSig software (Tucker Davis Technologies, Alachua, FL, USA).

The response threshold was determined at each frequency as the minimal sound pressure evoking a noticeable potential peak in the expected time window of the recorded signal. The average values per group were calculated (mean ± SD) and plotted in audiograms.

ABR wave amplitudes were analyzed using ABR response curves, obtained in response to a click stimulation of 90 dB SPL. ABR wave amplitudes were determined as the distance between the starting negative peak and the following positive peak for wave I and wave IV, ratios of wave I to wave IV amplitude were calculated and compared between the groups.

### 4.4. Recording of the Neuronal Activity in the IC

Mouse IC was accessed through small holes drilled into the skull and neuronal activity was recorded using a 16-channel, single shank probe (NeuroNexus Technologies, Ann Arbor, MI, USA), with 100 m between the electrode spots. Acoustic stimuli were generated with a TDT System III using the RP 2.1 Enhanced Real-Time Processor (Tucker Davis Technologies, Alachua, FL, USA), and delivered in free-field conditions via a two-way loudspeaker system (Jamo woofer and SEAS T25CF 002 tweeter), which was placed 70 cm in front of the animal’s head. On average, five electrode penetrations per animal were made. The first penetration was made in the middle of the IC.

The signal obtained from the electrode was amplified 10,000 times, band-pass filtered over the range of 300 Hz to 10 kHz, and processed by a TDT System III (Tucker Davis Technologies, Alachua, FL, USA), using an RX5-2 Pentusa Base Station. Using BrainWare software (v. 8.12, Jan Schnupp, Oxford University, UK), individual spikes from the recorded signal were isolated on-line on the basis of amplitude discrimination; the threshold was set adaptively according to the level of noise in the individual channels. Subsequent discrimination of spikes from the recorded data and their sorting according to the amplitudes of the first positive and negative peaks were performed off-line. The recorded data was processed and analyzed using custom made software based on MATLAB.

The rate-intensity functions (RIF) were constructed based on neuronal responses to broadband noise (BBN) bursts of variable intensity (from 0 to 90 dB SPL, 10 dB steps, 50 repetitions), and were then used to determine the response threshold, maximal response magnitude and dynamic range of the RIFs as previously described [48]). Post-stimulus time histograms to BBN stimuli at 80 dB SPL were used to obtain minimum first-spike latency (FSL), as the amount of time between the onset of sound presentation and the appearance of first spikes of neuronal responses. The spontaneous firing rate of each neuron was determined from the post-stimulus time histogram, as the number of spikes in the time interval between 200 and 300 ms after the presentation of BBN stimulus 10 dB above the threshold.

The tuning properties of individual IC neurons were evaluated using frequency-intensity mapping as described [49]. Briefly, the excitatory response areas were obtained via recording responses to pure tone bursts (100 ms in duration, 5 ms rise/fall times) with variable frequency (1/8 octave step) and intensity (5 dB step), presented in a random order. A two-dimensional matrix with elements corresponding to the response magnitudes in the respective frequency-intensity points was created, and then converted to a smooth function of two variables, frequency and intensity, via cubic smoothing spline interpolation. The following parameters of interest were then extracted from the resulting smooth function: (i) the excitatory response threshold, the lowest stimulus intensity that excited the neuron, measured in dB SPL; (ii) the characteristic frequency (CF), the frequency with the minimal response threshold, measured in kHz; and (iii) the bandwidth of the excitatory area 10 dB above the excitatory threshold, measured by quality factor Q_10_. Quality factor Q_10_ is a common measure of frequency selectivity that is reciprocally related to the excitatory area bandwidth (the higher the Q_10_, the sharper the response), defined as Q_10_ = CF/bandwidth.

To detect the inhibitory areas, a two-tone stimulation was employed [49]. A simultaneous presentation of the pure tone at the neuron’s CF fixed 10 dB above the threshold and pure tone bursts of variable frequency and intensity, analogous to those used for the excitatory area mapping, allowed us to create a two-dimensional matrix with distinguishable excitatory, inhibited and non-inhibited areas. Spike rates of responses in non-inhibited and inhibited areas were determined. Inhibitory strength was calculated as the ratio of spike numbers per stimulus in inhibited areas to those in non-inhibitory areas. Low- and high-frequency inhibition bands were analyzed separately.

To test the temporal processing ability of individual IC neurons, the animals were stimulated with click trains (positive rectangular pulses with duration of 100 ms) at an intensity of 70 dB SPL with different inter-click intervals (ICI), ranging from 2 to 100 ms. Each stimulus consisted of five clicks with a given ICI. The click trains with different ICIs were presented randomly, each stimulus occurring 30 times. The ability of neurons to synchronize with the click trains was evaluated using vector strength (VS); an example is described in [50]. The VS value is a number between 0 and 1, expressing the degree of synchronization of neuronal spike trains with a periodic stimulus. A higher VS indicates better synchronization ability.

To assess neuronal sensitivity to upward and downward frequency modulation, frequency modulated sweeps (FM) were used (1 sec duration, linear sweep from 2 kHz to 40 kHz and back, 60 dB SPL). Subsequently, the direction selectivity index (DSI) was calculated using the formula:(1)$$\mathrm{DSI}=\frac{\left(R_{u}-R_{d}\right)}{\left(R_{u}+R_{d}\right)}\;,$$
where $R_{u}$ is the number of spikes recorded from the IC neuron in response to an upward FM sweep, and $R_{d}$ is the number of spikes recorded from the same neuron in response to a downward sweep [51].

### 4.5. Immunohistochemistry and Morphological Analysis

The animals were given an overdose of thiopental (100 mg/kg) and perfused intracardially with 4% paraformaldehyde (Sigma-Aldrich Corp., St. Louis, MO, USA). The brain was removed, postfixed in the same solution for 48 h, sectioned into 80 µm coronal vibratome sections or cryoprotected overnight in 30% sucrose, and sectioned into 40 µm thick coronal sections on a freezing microtome (Leica Biosystems, Buffalo Grove, IL, USA). Three brains per group were analyzed. The brain regions were identified using a mouse brain atlas of Paxinos and Franklin [52], and brain slices between bregma −4.96 and −5.2 mm were chosen for further processing. Free-floating sections were blocked in 10% goat serum for 1 h, and then incubated for 48 h in anti-calretinin (rabbit, Millipore Sigma-Aldrich Corp., St. Louis, MO, USA, 1:1000), anti-calbindin (mouse, Merck Sigma-Aldrich Corp., St. Louis, MO, USA, 1:250), anti-NeuN (rabbit, Abcam, Cambridge, UK, 1:500), and anti-parvalbumin (rabbit, Sigma-Aldrich Corp., St. Louis, MO, USA, 1:1000) antibodies diluted in 0.3% triton with 2% goat serum. Further, sections were treated with a mixture of secondary antibodies: anti-mouse Alexa 594 and anti-rabbit Alexa 488 for 2 h. After washing, the sections were mounted onto slides, counterstained with DAPI, and cover-slipped. The preparations were examined, and images were captured using Zeiss 510 DUO laser confocal microscope (Oberkochen, Germany) with ×40 Plan-Apochromat oil immersion objectives (numerical aperture 1.4).

The area of the IC was established in 80 µm thick coronal brain sections from vibratome (Leica Biosystems, Buffalo Grove, IL, USA). Five adjacent sections containing the largest area of the IC per animal were measured. The areas of the left and right IC were determined in each section using ImageJ software. For the cell quantification, two 40 µm thick coronal brain sections per animal were used. Tiled images of the area of the whole colliculus were obtained at different levels through thickness of the section with a 7 µm step and processed as z-stacks using ImageJ software. Image processing and quantification analysis were performed blindly by the investigator. The cells were counted separately in central (CIC), dorsal (DIC), and external (EIC) nuclei of the IC identified according to a mouse brain atlas of Paxinos and Franklin [52]. ImageJ plug-in “Grid” was used to overlay the area of the IC with a grid of 200 µm × 200 µm squares. Using the “Cell counter” plugin, cells were counted in every second square of the grid. In this way, the cells were counted in approximately 50% of the area of the colliculus on each brain section. The cell density was calculated using the following formula:(2)$$D=\frac{N_{cells}}{N_{sq\;}\cdot \;A_{sq}\;\cdot \;T_{sl}}$$
where *D* is cell density, *N_cells_* is number of cells obtained during counting, *N_sq_* is the number of grid squares in which the cells were counted, *A_sq_* is grid square area, and *T_sl_* is the brain slice thickness.

### 4.6. Gene Expression Analysis

Total RNA was isolated from both left and right ICs of 1-month old mice, using TRI Reagent (Sigma-Aldrich Corp., St. Louis, MO, USA, T9424). We used eight IC samples for both *Tg* and WT. RNA samples (1 μg) were subjected to reverse transcription (RT), as described [22]. Following RT, quantitative qPCR was performed with initial activation at 95 °C for 120 s, followed by 40 cycles at 95 °C for 15 s, 60 °C for 30 s, and 72 °C for 30 s, using the CFX384™ Real-Time PCR Detection System (Bio-Rad Laboratories, Hercules, CA, USA). The primer sequences obtained from (pga.mgh.harvard.edu/primerbank/, accessed on 17 March 2021) are listed in Table 1. The relative mRNA expression was calculated using the –ΔΔCq method, with *Hprt1* as a reference gene [53].

### 4.7. Statistical Analysis

Data are presented as the mean ± standard deviation (SD) for values with normal distributions or the median (Mdn) with interquartile ranges and extremes for values with non-normal distributions. For statistical analysis, GraphPad Prism software (version 8.0.0 for Windows, GraphPad Software, San Diego, CA, USA, www.graphpad.com, accessed on 19 April 2021) was used. To assess the differences in mean or median values between the groups, repeated two-way ANOVA with Uncorrected Fisher’s LSD post-hoc test, unpaired *t*-tests, or Mann-Whitney tests were used.

## Figures and Tables

**Figure 1 ijms-22-04507-f001:**
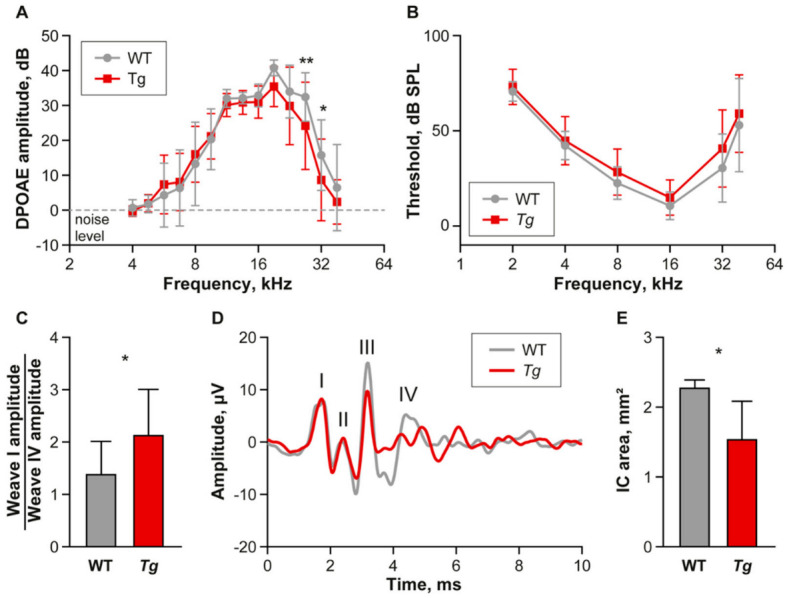
Hearing function of wild type and *Tg* animals. (**A**) Distortion product otoacoustic emissions are significantly impaired in high frequencies of *Tg* mice. Mean ± SD, repeated measures two-way ANOVA, Uncorrected Fisher’s LSD post-hoc test, * *p* < 0.05, ** *p* < 0.01. (**B**) ABR thresholds are not different between the groups. Mean ± SD, repeated measures two-way ANOVA. (**C**) Ratio of wave I amplitude to wave IV amplitude is increased in transgenic animals compared to wild type. Unpaired *t*-test, * *p* < 0.05. (**D**) Example of ABR curve showing impaired wave IV in transgenic animal compared to wild type. (**E**) Comparison of IC area in WT and *Tg* brains.

**Figure 2 ijms-22-04507-f002:**
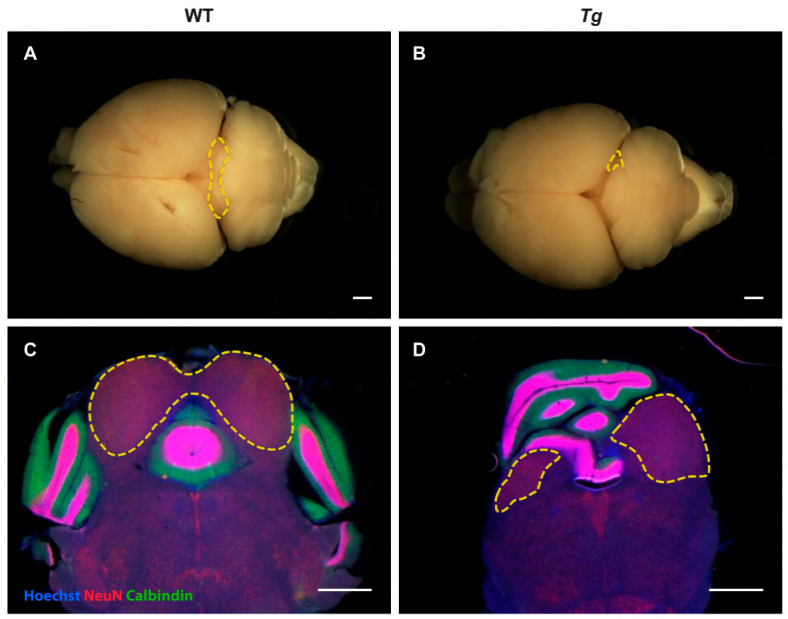
Size and position of the inferior colliculus in WT and *Tg* brain (animals with highly expressed phenotype). (**A**,**B**) Simple inspection of the brain surface from the dorsal view reveals smaller size of inferior colliculus in *Tg* brain (delineated in **B**) compared to WT brain (delineated in **A**). (**C**,**D**) Same finding confirmed using stained coronal brain sections. Scale bar 1 mm.

**Figure 3 ijms-22-04507-f003:**
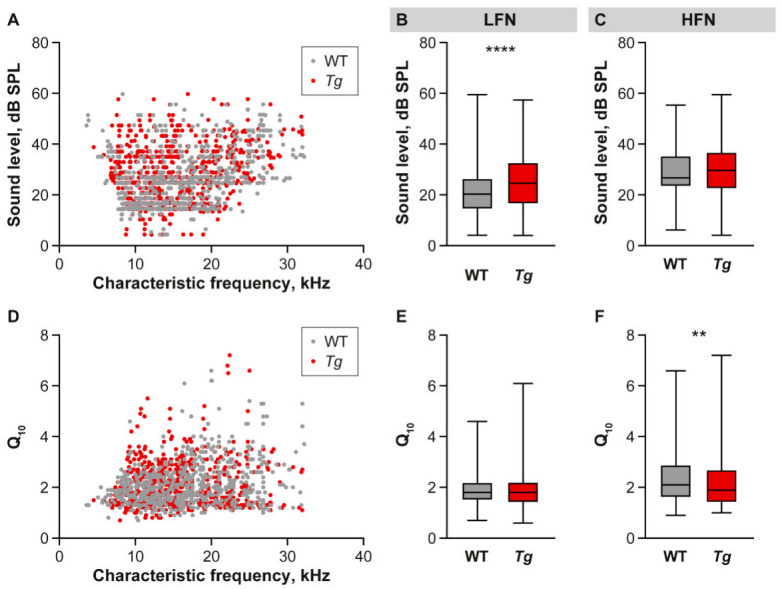
Characteristics of IC neuronal responses to pure tones. (**A**) Scatter plot showing thresholds of all measured neurons as a function of characteristic frequency in both groups of animals. (**B**,**C**) Box and whiskers plots showing distribution of neuronal thresholds separately in LFN and HFN in the WT and *Tg* IC. Median with interquartile range and extremes, Mann-Whitney test, **** *p* < 0.0001. (**D**) Scatter plot showing the quality factor of all measured IC neurons as a function of characteristic frequency in both groups of animals. (**E**,**F**) Box and whiskers plots showing distribution of neuronal quality factors separately in LFN and HFN. Median with interquartile range and extremes, Mann-Whitney test, ** *p* < 0.01.

**Figure 4 ijms-22-04507-f004:**
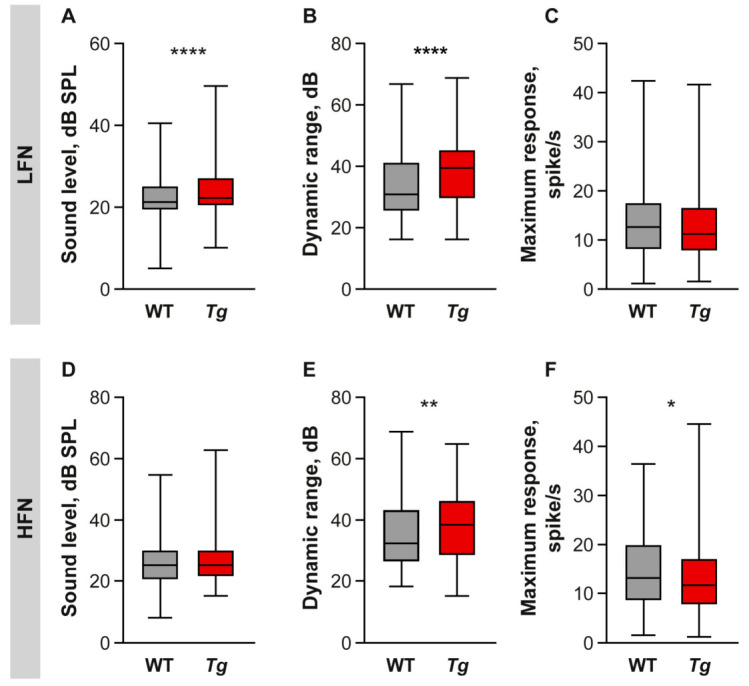
Characteristics of the rate-intensity functions measured using broad-band noise. (**A**–**F**) Box and whiskers plots depicting distribution of neuronal threshold (**A**,**D**), dynamic range (**B**,**E**) and maximum response (**C**,**F**) values in LFN (**A**–**C**) and HFN (**D**–**F**) of WT and *Tg* mice. Median with interquartile range and extremes, Mann–Whitney test, * *p* < 0.05, ** *p* < 0.01, **** *p* < 0.0001.

**Figure 5 ijms-22-04507-f005:**
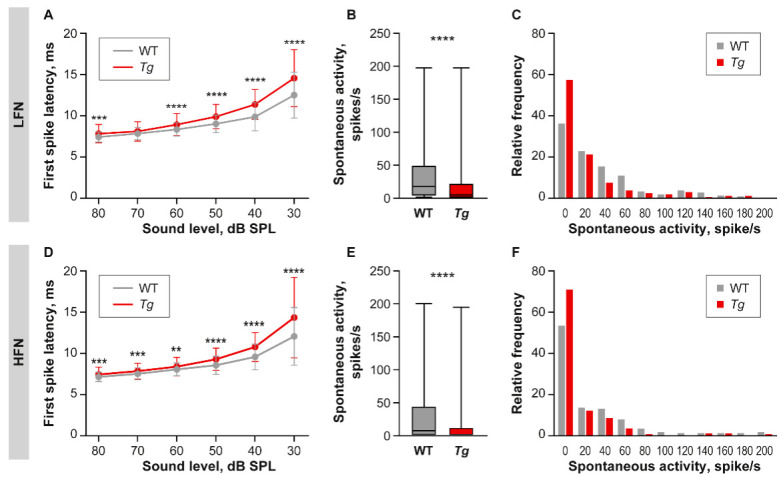
First spike latency and spontaneous activity. (**A**,**D**) FSL as a function of sound intensity for LFN (**A**) and HFN (**D**) in WT and *Tg* IC. Repeated measures two-way ANOVA, Uncorrected Fisher’s LSD post-hoc test, ** *p* < 0.01, *** < 0.001, **** *p* < 0.0001. (**B**,**E**) Box and whiskers plots showing distribution of spontaneous activity values in LFN (**B**) and HFN (**E**). Median with interquartile range and extremes, Mann-Whitney test, *** <0.001, **** *p* < 0.0001. (**C**,**F**) Histogram showing distribution of spontaneous activity values in LFN (**C**) and HFN (**F**).

**Figure 6 ijms-22-04507-f006:**
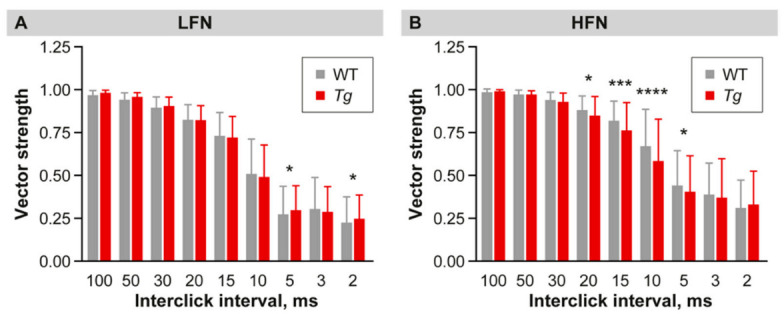
Precision of temporal coding. (**A**,**B**) Ability of LFN (**A**) and HFN (**B**) to synchronize with click trains with different inter-click intervals expressed using vector strength (VS) values. Mean ± SD, one-way ANOVA, Fisher’s LSD post-hoc test, * *p* < 0.05, *** <0.001, **** *p* < 0.0001.

**Figure 7 ijms-22-04507-f007:**
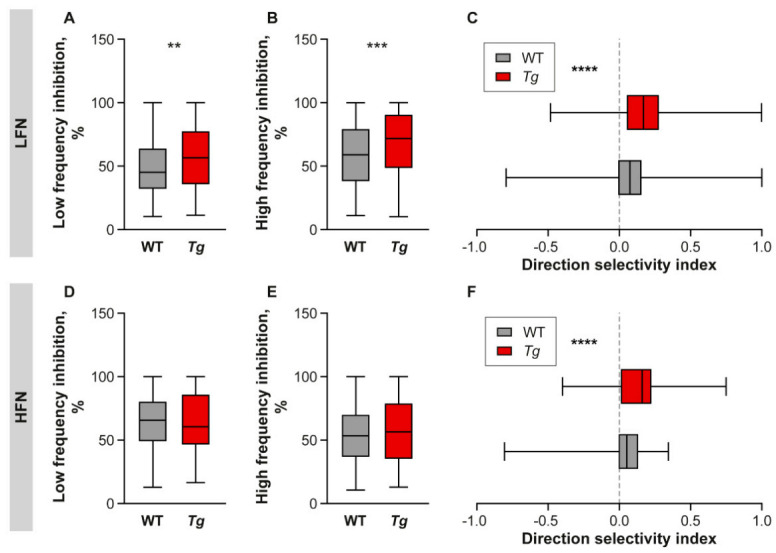
Two-tone inhibition and FM direction selectivity. (**A**,**B**,**D**,**E**) Comparison of low- (**A**,**D**) and high-frequency (**B**,**E**) sideband inhibition strength in LFN (**A**,**B**) and HFN (**D**,**E**) in WT and Tg IC. Median with interquartile range and extremes, Mann-Whitney test, ** *p* < 0.01, *** <0.001. (**C**,**F**) Frequency modulated stimulus direction selectivity of LFN (**C**) and HFN (**F**). Median with interquartile range and extremes, Mann-Whitney test **** *p* < 0.0001. HFN: high frequency neurons, LFN: low frequency neurons.

**Figure 8 ijms-22-04507-f008:**
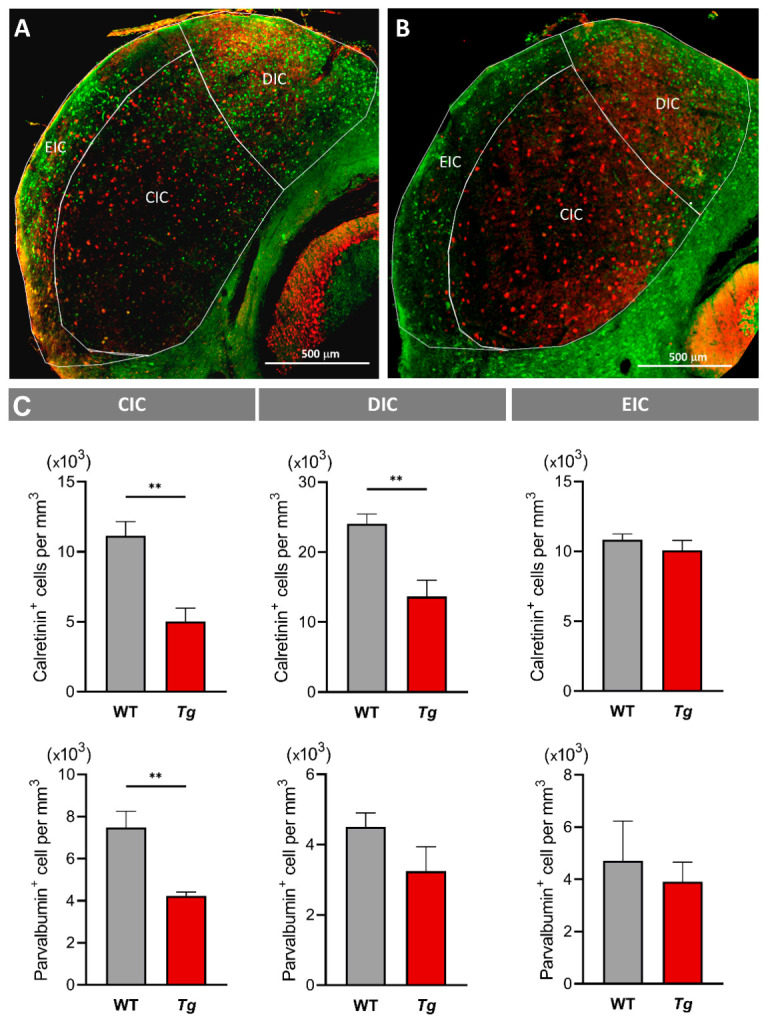
Calcium binding proteins in the IC. (**A**,**B**) Illustration of immunofluorescent staining of the coronal brain section with anti-calretinin (green) and anti-parvalbumin (red) staining in WT (**A**) and *Tg* (**B**) mice. Delineations show central (CIC), dorsal (DIC), and external (EIC) nuclei of the IC. (**C**) Number of calretinin and parvalbumin positive cells in the IC nuclei. Mean ± SD, unpaired *t*-test, ** *p* < 0.01.

**Figure 9 ijms-22-04507-f009:**
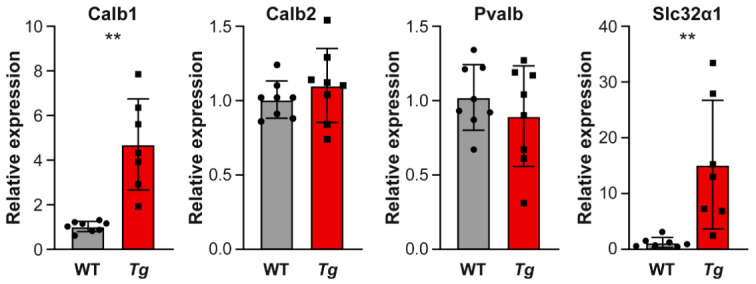
Calcium binding protein and inhibitory gene expression in the IC. Expression of mRNA of GABA and glycine transporter (*Slc32a1*) and calcium binding proteins calbindin (*Calb1*), calretinin (*Calb2*), parvalbumin (*Pvalb*) in the WT and *Tg* IC. Mean ± SD, unpaired *t*-test, ** *p* < 0.01.

**Table 1 ijms-22-04507-t001:** List of primers for qPCR.

*Hprt1 F*	GCTTGCTGGTGAAAAGGACCTCTCGAAG
*Hprt1 R*	CCCTGAAGTACTCATTATAGTCAAGGGCAT
*Calb1 F*	GATTGGAGCTATCACCGGAA
*Calb1 R*	TTCCTCGCAGGACTTCAGTT
*Calb2 F*	TGATGCTGACGGAAATGGGT
*Calb2 R*	CCCTTCCTTGCCTTCTCCAG
*Pvalb F*	ATCAAGAAGGCGATAGGAGCC
*Pvalb R*	GGCCAGAAGCGTCTTTGTT
*Slc32a1 F*	ACCTCCGTGTCCAACAAGTC
*Slc32a1 R*	CAAAGTCGAGATCGTCGCAGT

*Hprt1*, hypoxanthine guanine phosphoribosyl transferase 1; *Calb1*, calbindin1; *Calb2*, calbindin2 (synonyms: calretinin); *Pvalb*, parvalbumin; *Slc32a1*, solute carrier family 32 (GABA vesicular transporter).

## Data Availability

The data presented in this study are available in this article.

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
