# Peer review of "Overexpression of Isl1 under the Pax2 Promoter, Leads to Impaired Sound Processing and Increased Inhibition in the Inferior Colliculus"

_ijms, 2021, doi:10.3390/ijms22094507_

Round 1
Reviewer 1 Report
This manuscript demonstrates alterations to auditory morphology and physiology of the inferior colliculus in transgenic mice overexpressing Isl1. I have a few comments.
In the abstract, the significance of Isl1 should be briefly explained.
Regarding the reduction in volume of the IC in transgenic animals, from Figure 2, it also appears that the gross brain size is also smaller. If so, how does the IC reduction compare to the overall reduction in brain volume? Are other auditory structures similarly reduced in size? Also, how were the two adjacent 80um sections chosen for analysis?
Regarding the neuronal recordings in the IC, the authors report no difference in the distribution of frequency responses in transgenic animals, despite the morphological difference. It might be interesting to see a plot of best frequency responses as function of penetration depth, normalized for each animal. Also, were any of the recording sites marked with electrolytic lesions to assess which nucleus in the IC was being recorded?
The authors discuss the vagaries and limitation of the two-tone suppression method in assessing inhibitory changes in the IC. Although the discussion is well presented, it would have been preferred if the authors had undertaken experiments, e.g. pharmacological, to better assess this in the study.
Regarding quantification of parvalbumin and calretinin in Figure 8, the divisions shown for the analysis are very broad. The purported LFN region covers not only the low frequency part of the central nucleus, but also the entire dorsal nucleus and part of the lateral nucleus. Similarly, the HFN region also contains the lateral nucleus. I think this analysis needs to be redone to specify the calcium-binding expression to specific IC nuclei (i.e. central nucleus, dorsal nucleus, lateral nucleus).
Regarding the statistics, the error bars in several figures seem to indicate no statistical difference, where one is indicated.
Line 219: What is the commissural nucleus? Do you mean central nucleus of the IC?
I am not sure why the primer list needs to be separated in supplementary material. It is simple enough to include in the main text.
There are a few grammatical mistakes and odd word choices throughout the manuscript, which may benefit from review by a native English speaker.
Reviewer 2 Report
Comments to Authors
Manuscript number: ijms-1167820
Title: Overexpression of Isl1 under the Pax2 Promoter, Leads to Impaired Sound Processing and Increased Inhibition in the Inferior Colliculus
Chumak, Tetyana; Tothova, Diana; Filova, Iva; Bures, Zbynek; Popelar, Jiri; Pavlinkova, Gabriela; Syka, Josef
Overview and general recommendation:
The current study characterized sound processing in transgenic mice with Pax2 driven Isl1 overexpression, which show abnormalities in the inner ear, medial olivocochlear system, and auditory midbrain. The authors assessed how overexpression of Isl1 under Pax2 regulatory sequences, associated with abnormalities in the development of the mid-hindbrain region, affects sound processing by IC neurons. Especially they analyzed measures of inhibition in the IC, with focus of the role on inhibition in the IC.
I found the paper overall well written, clearly structured and the results are well described. In addition, the figures were clear and informative. I came away with only few questions and comments. I still recommend that a revision is warranted. I kindly ask that the authors specifically address each of my comments in their response.
Main comment:
My main comment is that it is not fully elaborated and clear in the text why exactly this specific mouse model was used? And why this model was used in order to answer what specific questions about processing in the auditory system? This comes out a bit unclear, and the manuscript could benefit if that was elaborated and motivated a bit more in the specific parts of the manuscript.
Comments:
Abstract
- Page 1, Line 15. First occurrence of Tg is not defined. Also using italic in the remaining text might lead to confusion with an italicized gene name. Here it is supposed to stands here for transgenic. In any way, at least it needs to be properly defined, because for some readers not familiar with gene nomenclature this might be confusing.
Keywords
- Page 1, Line 27. Why are there numbers behind each keyword? Is that an error?
Introduction
- Page 2, Line 1. What is lateral mmnucleus? Provide full name.
Results:
- Page 3, Line 88. Why was wave IV chosen and not wave V? What specific parts of the auditory brainstem is wave IV thought to represent? Elaborate. Provide some references.
- Page 4, Line 103. Can you comment on the animals in which the IC was so reduced that no probe recordings were made? Why were there differences between animals, and how pronounced were they? How many animals were excluded (provide numbers in methods)? Do you assume that general responses were similar in them to the animals, which were not excluded? Were the ABR waves in these animals different, and were these animals included in the ABR analyses, but not included in the microelectrode recordings? Did these animals have similar hearing thresholds? Explain and specify.
Methods:
- Page 12, Line 376. How was the recording position chosen? In the middle of the IC? How many electrode penetrations per each animal were measured? Specify.
- Page 12, Line 383. How was the multiunit activity extracted from the raw signal? What threshold was used? Specify.
